# Use of a decision support system for benchmarking analysis and organizational improvement of regional mental health care: Efficiency, stability and entropy assessment of the mental health ecosystem of Gipuzkoa (Basque Country, Spain)

**Carlos R. García-Alonso**[1], **Nerea Almeda**[2]*, **José A. Salinas-Pérez**[1], **Mencía R. Gutiérrez-Colosía**[2], **Álvaro Iruin-Sanz**[3], **Luis Salvador-Carulla**[4]

**1** Department of Quantitative Methods, Universidad Loyola Andalucía, Seville, Spain, **2** Department of Psychology, Universidad Loyola Andalucía, Seville, Spain, **3** Instituto Biodonostia, Red de Salud Mental Extrahospitalaria de Gipuzkoa, Donostia-San Sebastián, Spain, **4** Health Research Institute, Faculty of Health, University of Canberra, Canberra, Australia

\* nmalmeda@uloyola.es

## Abstract

Decision support systems are appropriate tools for guiding policymaking processes, especially in mental health (MH), where care provision should be delivered in a balanced and integrated way. This study aims to develop an analytical process for (i) assessing the performance of an MH ecosystem and (ii) identifying benchmark and target-for-improvement catchment areas. MH provision (inpatient, day and outpatient types of care) was analysed in the Mental Health Network of Gipuzkoa (Osakidetza, Basque Country, Spain) using a decision support system that integrated data envelopment analysis, Monte Carlo simulation and artificial intelligence. The unit of analysis was the 13 catchment areas defined by a reference MH centre. MH ecosystem performance was assessed by the following indicators: relative technical efficiency, stability and entropy to guide organizational interventions. Globally, the MH system of Gipuzkoa showed high efficiency scores in each main type of care (inpatient, day and outpatient), but it can be considered unstable (small changes can have relevant impacts on MH provision and performance). Both benchmark and target-for-improvement areas were identified and described. This article provides a guide for evidence-informed decision-making and policy design to improve the continuity of MH care after inpatient discharges. The findings show that it is crucial to design interventions and strategies (i) considering the characteristics of the area to be improved and (ii) assessing the potential impact on the performance of the global MH care ecosystem. For performance improvement, it is recommended to reduce admissions and readmissions for inpatient care, increase workforce capacity and utilization of day care services and increase the availability of outpatient care services.

**Data Availability Statement:** All Mental Health Network of Gipuzkoa Dataset (2015) for Benchmark Analysis files are available from the Dryad database (accession number(s): https://datadryad.org/stash/share/cFD5jA5F3odtAhZqUARiPBTo_yj8lKvQ9GWAGrauPnE).

**Funding:** This study was funded by the Instituto de Salud Carlos III (grant number: PI18/01521; author who received the award: CRGA) https://www.isciii.es/Paginas/Inicio.aspx and Junta de Andalucía con Fondos FEDER (Unión Europea) (grant number: Y18-RE-0022; author who received the award: CRGA) https://www.juntadeandalucia.es/economiaconocimientoempresasyuniversidad/fondoseuropeosenandalucia/feder. There was no additional external funding received for this study. The funders had no role in study design, data collection and analysis, decision to publish, or preparation of the manuscript.

**Competing interests:** The authors have declared that no competing interests exist.

# Introduction

The shift of care provision from institutions to the community started around the 1950s and culminated with the development of the extended balance of care model [1,2]. However, despite great efforts [3], the shift of resources from institutions to the community is a dynamic and complex goal and not a reality worldwide. Currently, in Europe, (i) more than one million persons are institutionalized, (ii) the number of persons living in institutions has not decreased over the last ten years, (iii) people stay longer in institutions because of the lack of placement capacity in community services and (iv) persons with disabilities and complex mental health (MH) needs live in institutions instead of the community [4].

In 2002, Thornicroft and Tansella developed the basic model of balanced MH care provision [5]. They identified primary care services and outpatient care services, including mobile and nonmobile services; day care services, such as vocational and rehabilitation services; links with other types of services, such as social services or nongovernmental organizations; and acute inpatient care services and long-term residential care services. The balance is aimed at optimizing care provision by providing as much community care as possible and as little hospital care as possible. Nevertheless, community care cannot work alone, and changes should be gradual and incremental, considering that there will always be a need for a minimum number of beds in hospitals [6]. In 2013, the same authors developed a new version called the balanced MH care model, which defined the main types of MH care that countries should deliver according to their income level [7]. According to this model, MH ecosystems should include different combinations of primary care services, general MH services and specialized MH services (e.g., outpatient care clinics, community MH teams, early intervention teams). Among the new types of services, the authors highlighted day care services [8], assertive community treatment [9], early intervention in psychosis [10], crisis resolution teams [11,12] and supported accommodation services [13] to provide specialized MH care in the community meeting users' needs while trying to promote autonomy and independence among people suffering from mental disorders.

From a health ecosystem perspective [14], the balance of care model should be considered an intrinsically system-based approach. Therefore, it does not intend to achieve symmetry between hospital and community services or to compare evidence of one service against the other. In contrast, it aims to find an optimal balance for improving care efficiency that should be quantified at every level: micro (individual services), meso (health districts) and macro (regions or countries) [6]. As proposed by Rosen et al. [6], we should "move from a binary "seesaw" representation of the balance of care to a multidimensional model like Alexander Calder's mobiles".

Currently, planning MH care requires improved management methods to integrate different types of care and guarantee appropriate care provision. Recently, Bouras, Ikkos and Craig [15,16] developed the meta-community MH care model, which proposed a theoretical framework including the lessons learned from deinstitutionalization and new insights such as primary care relevance, the necessity of accessible, accessible and coordinated MH services and finally, the participation of users to reduce stigma.

Decision support systems (DSSs) are useful computer-based tools for guiding decision-making processes, particularly in complex ecosystems such as MH care provision research and planning [17]. Two recent reviews show that these tools have been widely used over the years to support clinicians' decision-making [18,19]. Their use in MH care has been lower than in other areas of medicine. This use has included policy design [20,21], mood assessment [22] or risk evaluation of violent reoffending [23]. DSSs integrate statistical and operational techniques, and previous works [21,24] have shown the utility of indicators such as relative

technical efficiency (RTE), statistical stability and entropy to analyse MH services and ecosystem performance [25,26]. In health care, RTE has been frequently assessed over the years [27], but its use for the performance assessment of MH services has been scarcer [28]. One of the most useful applications of DSSs for MH provision management is identifying both benchmark and target-for-improvement decision units. The former can offer ideas about best managerial practices, and by comparison, decision-makers could identify improvement areas for the latter. Nevertheless, to assess the indicators, DSSs must integrate formalized expert knowledge from the mentioned MH care models as a guide for analysis [29].

This paper aims to develop an analytical process for (i) assessing the performance of a regional MH ecosystem (the Mental Health Network of Gipuzkoa, Basque Country, Spain) structured in reference catchment areas (decision-making units) that provide inpatient acute care, core health day care and outpatient care and (ii) identifying benchmark and target-for-improvement catchment areas. Selecting and assessing key variables must be modified to improve MH care performance. These variables and their potential modifications can be used to design potential organizational interventions and policies. According to this, the paper is organised in the following sections: (i) Methods, to firstly describe the real system structure and the available dataset (variables) and secondly to set the basic indicator definitions and briefly analyse the usability of the DSS in supporting the analysis of real interventions and policies, (ii) Results, analysing the MH system performance (relative technical efficiency, stability and entropy) and identifying potential interventions on "target for improvement areas" considering the structure of "benchmarking" ones and, finally, (iii) this paper ends with the discussion and conclusion sections.

## Methods

This study is a demonstration study of the usability of a novel DSS for guiding evidence-informed regional planning in MH care. It adopts the perspective of a public agency (Mental Health Network of Gipuzkoa, Basque Country, Spain).

MH services were classified using the DESDE-LTC codification system [30] into three main types of care: residential, core health day and outpatient care. The use of this classification prevents service non-commensurability and terminological diversity biases and thus facilitates comparisons across territories and countries [31]. In the Gipuzkoa MH ecosystem, non-acute hospital care is assumed to be core health day and outpatient care.

### Study area and data

Basque Country is an Autonomous Community in Spain that has complete control over its health care provision. The Mental Health Network of Gipuzkoa (one of the three historical territories of the Autonomous Community) manages its community and rehabilitation MH services ($\cong$774,700 residents). These services include outpatient and core health day care facilities closely related to both hospital services (inpatient) and the social care network for severe cases. The Mental Health Network of Gipuzkoa is divided into 13 catchment areas, all of which have a mental health centre, considered as a reference, with a distinct orientation to outpatient care (some of them also include core health day care facilities).

Each catchment area was described by 25 variables: service availability, placement capacity and workforce capacity. Most variable values were expressed in rates per 100,000 inhabitants (adults, over 18 years old). Utilization and performance variables were selected to collect information about service use, treated prevalence and incidence. Hospital discharges and readmissions were obtained from the 2015 MH utilization database (Health care Dashboard, Mental Health Network of Gipuzkoa). The structural variables (resource availability) were considered

the input set, and utilization and performance variables (outcome proxies) were considered the output set (Table 1).

To describe MH provision from the main care perspectives, selected variables were combined in three scenarios: acute hospital care, core health day care and non-acute outpatient care. Following this strategy, the results from the DSS offered a set of three performance scores for each catchment area. Following this analytical strategy, this study focuses its attention on basic MH care services excluding social, education, and other non-care services.

The inputs for the inpatient acute hospital care scenario were the number of available services (rate per 100,000 inhabitants), number of beds (rate), number of psychiatrists (rate), number of psychologists and nurses (rate) and total number of professionals (rate), and the outputs were discharges (number of users), readmissions (number of users) and average length of stay (days). For the acute and non-acute core health day care scenario, the inputs were the number of available services (rate), number of psychiatrists (rate), number of psychologists and nurses (rate), total number of professionals (rate) and placement capacity (rate), and the outputs were the utilization of acute core health day care health services (number of places) and utilization of non-acute day care core health services (number of places). Finally, for the non-acute outpatient care scenario, the inputs were the number of available services (rate), number of psychiatrists (rate), number of psychologists (rate), number of nurses (rate) and total number of professionals (rate), and the outputs were the incidence (number of users), prevalence (number of users) and contacts (number of users) (Table 1). Data set is available at the Dryad digital repository (https://datadryad.org/stash/share/cFD5jA5F3odtAhZqUARiPBTo_yj8lKvQ9GWAGrauPnE) and processed datasets that feed the DSS are available upon request.

All of the catchment areas have their own MH centre (all of them have one outpatient care facility with specific characteristics), but acute inpatient care (one hospital physically located in Donostia: areas 2, 6 and 9) and core health day care (areas 1, 2, 5–11 and 13) facilities have wider influence areas (they are specifically located in selected catchment areas and users must go there when needed) to provide MH care to the whole ecosystem. For these two main MH types (inpatient scenario S1 and core health day care scenario S2), input and output scores were calculated proportionally according to (i) the catchment area population, (ii) the area of influence of the corresponding facilities and (iii) the geographical distance.

### Indicator definitions

The methodology developed in this paper combined qualitative and quantitative processes for assessing ecosystem performance (from three perspectives) and the proposed and potential

**Table 1. Scenarios and variables identified by experts (DESDE-LTC codes are also included).**

| Scenarios | Inputs | Outputs |
|---|---|---|
| S1: Acute hospital care scenario (code: R2) | 1. Availability<br>2. Beds<br>3. Workforce capacity (psychiatrists, nurses, psychologists and total number of professionals) | 1. Readmissions<br>2. Length of stay<br>3. Discharges |
| S2: Core health day care (codes: D1 and D4.1) | 1. Availability<br>2. Placement capacity<br>3. Workforce capacity (psychiatrists, nurses, psychologists and total number of professionals) | 1. Utilization |
| S3: Outpatient care (codes: O8-O10) | 1. Availability<br>2. Workforce capacity (psychiatrists, nurses, psychologists and total number of professionals) | 1. Prevalence<br>2. Incidence<br>3. Contacts |

**Table 2. Glossary of key terms and indicators for assessing ecosystem performance.**

| Key terms | | Definitions |
|---|---|---|
| Organizational interventions | Structural | Specific action whose main impact is on the structure of MH care provision (e.g., open a new outpatient facility). |
| | Administrative | Specific action based on modifying the administrative structure of MH care provision (e.g., modify the size of a catchment area). |
| | Procedural | Specific action that modifies a procedure or a process (e.g., to modify a pathway or care). |
| Systems Performance Indicators | Relative Technical Efficiency (RTE) | Indicator that assesses the relationship among the inputs used and outputs produced by combining them in a group of comparable decision-making units. It can be input or output oriented. Data Envelopment Analysis (DEA) is a set of robust techniques for assessing RTE. |
| | Stability | Indicator that assesses the potential impact of variable value changes on, for example, the RTE. High values of stability indicate that changes in variable values do not significantly affect the results (e.g., the ecosystem performance). Low stability scores indicate that small changes in variable values can have a great impact (positive or negative) on the results. |
| | Entropy | Indicator that assesses the level of ecosystem heterogeneity or disorganization. High entropy scores indicate that the ecosystem is highly disorganized or has many tailormade strategies (heterogeneous management). |
| Clusters (branches) of Main Types of Care (DESDE-LTC) | Residential | Acute care, 24 h physician cover, e.g., units in general hospitals and psychiatric hospitals. |
| | | Non-acute care, 24 h physician cover, e.g., rehabilitation units and nursing homes. |
| | Core Health Day Care | Acute, e.g., day hospitals. |
| | | Non-acute, non-work structured, health-related care, e.g., day care centres. |
| | Outpatient Non-Acute Care | Non-acute, health-related care, e.g., community mental health teams and outpatient psychiatric clinics. |

interventions [32]. The DSS was used to evaluate the situation of acute hospital care (S1), core health day care (S2) and outpatient care (S3). This tool included three composite ecosystem indicators: RTE (operational), stability (statistical) and entropy (statistical) (Table 2).

Regarding the ecosystem performance indicators for analysing a MH system, the RTE assesses the relationship between the inputs consumed and the outputs produced for a set of comparable decision-making units (DMUs, catchment area in this paper) [33]. A DMU is efficient when RTE is 1 and completely inefficient when it is 0. In this paper, input-oriented data envelopment analysis (DEA) with variable returns to scale was used to assess RTE [34]. This technique tries to minimize the number of inputs by maintaining a constant current number of outputs. If this situation is feasible, the DMU is not efficient (RTE less than 1). In contrast, the DMU is weakly efficient or efficient when the RTE is 1 (for a weakly efficient DMU, the sum of the slacks is not zero). A Monte Carlo simulation engine was integrated in the DSS to incorporate randomness in RTE assessment (500 simulations were carried out for each scenario). The resulting RTE scores can be statistically analysed to determine the basic RTE scores in terms of probability, stability and entropy.

Stability assesses the potential impact of variable value changes on the RTE over a [0, 100] range [21]. The stability score is 0% where the DMU and/or the whole ecosystem is completely unstable. This score implies that small changes in variable values produce very large changes in ETR scores, and these changes can be positive or negative. In contrast, they are completely stable when (stability equal to 100%) changes in variable values cannot significantly modify RTE scores. If stability is low, the performance of the MH ecosystem can react in an unpredictable way when an intervention is implemented. Therefore, decision-makers usually seek to increase both DMU and ecosystem stabilities.

The ecosystem management style was assessed by Shannon's entropy over a [0, 100] range [35]. Its management is completely homogeneous when the entropy score is 0% on its maximum feasible value, meaning that all the catchment areas are managed identically (all the ecosystem DMUs show a similar structure and managerial procedures, usually following a specific

policy). The management style is completely heterogeneous when the entropy score is 100%, which means that each area has its own tailormade management procedures.

Entropy is also associated with the variability assessment of the variable values for each DMU. In this paper, the Monte Carlo simulation engine has exhibits variability under control. High levels of DMU entropy make the systems extremely responsive to environmental changes.

## Potential usability of the DSS in real complex systems

DSSs systems are considered appropriate tools for guiding operational (basically resource allocation and use) evidence-informed planning and management. DSSs allow decision makers and policy makers to increase their knowledge about how the organization of the ecosystem (divided in comparable organizational units) manages scarce resources in order to produce mensurable and positive results, in the end: the ecosystem performance. Commonly, decisions and policymaking are based on the manager's experience, external opinions, historical facts, etc., and they need new evidence-based tools for increasing their knowledge when organizational interventions or policies must be assessed in advance. The DSS proposed in this paper allows them to have an objective assessment about the ecosystem performance including relative technical efficiency, stability and entropy, everything from different points of view. In addition, this computer-based tool can identify new improvement strategies once the proposed changes had been potentially taken in the real ecosystem, aiming at improving MH care provision.

## Results

### Relative technical efficiency: Resource management in the selected scenarios

All analysed MH areas have a high $\overline{RTE}$ (RTE on average, Table 3). This finding is especially relevant in the acute hospital (S1) and outpatient care (S3) scenarios and highlights the influence of low structural nuances in the management of MH care provision. The core health day care (S2) scenario shows more relevant differences and has a low $\overline{RTE}$ (Table 3). Here, it must be considered that some core health day care facilities are integrated in their respective MH centres (specialized in outpatient care, S3), as happens in areas 8 and 10. This characteristic makes the precise identification of core health day care inputs and outputs difficult.

In the acute hospital care (S1) scenario, areas 1, 3, 9 and 13 achieved the highest $\overline{RTE}$. They have the best input/output balance in the ecosystem, and they are the benchmarks. In contrast, areas 7 and 12 had the lowest $\overline{RTE}$ scores (Table 3), and therefore, they could be considered a priority for planning (target-for-improvement areas).

The core health day care (S2) scenario included both acute and non-acute health-related day care. Here, area number 11 achieved the best $\overline{RTE}$ (Table 3). Additionally, areas 3, 4 and 13 also achieved high $\overline{RTE}$ scores. In this scenario, areas 11 and 13 could be selected as the benchmarks. Areas 1, 7, 8 and 10 showed the lowest performance and are the target-for-improvement areas (Table 1). In the last two areas, 8 and 10, day care facilities are integrated into MH centres. As stated before, here, the availability and utilization of the corresponding services could be slightly underestimated because of the method selected for calculating their input and output scores (see the *Study area and data* section).

The outpatient care (S3) scenario included both non-acute and non-mobile care. In this case, areas 3, 4 and 5 achieved the best $\overline{RTE}$ (they are the benchmarks), while areas 1 and 2 showed the lowest performance and are thus target-for-improvement areas (Table 3).

**Table 3. Relative technical efficiency on average, RTE error on average and the probability of having an RTE greater than 0.75.** Darker shading indicates worse scores.

| Area | Indicator | Acute hospital care scenario | Day care scenario | Outpatient care scenario |
|---|---|---|---|---|
| **1** | $\overline{RTE}^{(1)}$ | 0.940 | 0.450 | 0.490 |
| | $P(RTE>0.75)$ | 1.000 | 0.000 | 0.000 |
| | Error $(\overline{\varepsilon})^{(2)}$ | 1.000 | 1.000 | 1.000 |
| **2** | $\overline{RTE}^{(1)}$ | 0.900 | 0.580 | 0.520 |
| | $P(RTE>0.75)$ | 1.000 | 0.000 | 0.000 |
| | Error $(\overline{\varepsilon})^{(2)}$ | 0.490 | 0.240 | 1.910 |
| **3** | $\overline{RTE}^{(1)}$ | 0.960 | 0.900 | 0.940 |
| | $P(RTE>0.75)$ | 1.000 | 1.000 | 0.996 |
| | Error $(\overline{\varepsilon})^{(2)}$ | 0.480 | 0.850 | 0.430 |
| **4** | $\overline{RTE}^{(1)}$ | 0.910 | 0.900 | 0.940 |
| | $P(RTE>0.75)$ | 0.998 | 1.000 | 1.000 |
| | Error $(\overline{\varepsilon})^{(2)}$ | 0.170 | 0.350 | 0.530 |
| **5** | $\overline{RTE}^{(1)}$ | 0.900 | 0.500 | 0.960 |
| | $P(RTE>0.75)$ | 1.000 | 0.000 | 1.000 |
| | Error $(\overline{\varepsilon})^{(2)}$ | 0.300 | 0.070 | 0.460 |
| **6** | $\overline{RTE}^{(1)}$ | 0.880 | 0.510 | 0.720 |
| | $P(RTE>0.75)$ | 0.998 | 0.000 | 0.436 |
| | Error $(\overline{\varepsilon})^{(2)}$ | 0.520 | 0.090 | 0.980 |
| **7** | $\overline{RTE}^{(1)}$ | 0.290 | 0.430 | 0.920 |
| | $P(RTE>0.75)$ | 0.000 | 0.000 | 1.000 |
| | Error $(\overline{\varepsilon})^{(2)}$ | 1.160 | 0.270 | 0.440 |
| **8** | $\overline{RTE}^{(1)}$ | 0.780 | 0.410 | 0.750 |
| | $P(RTE>0.75)$ | 0.782 | 0.000 | 0.550 |
| | Error $(\overline{\varepsilon})^{(2)}$ | 0.720 | 0.160 | 1.120 |
| **9** | $\overline{RTE}^{(1)}$ | 0.960 | 0.520 | 0.690 |
| | $P(RTE>0.75)$ | 1.000 | 0.000 | 0.162 |
| | Error $(\overline{\varepsilon})^{(2)}$ | 0.720 | 0.160 | 1.120 |
| **10** | $\overline{RTE}^{(1)}$ | 0.840 | 0.430 | 0.910 |
| | $P(RTE>0.75)$ | 0.904 | 0.000 | 0.910 |
| | Error $(\overline{\varepsilon})^{(2)}$ | 1.440 | 0.190 | 0.750 |
| **11** | $\overline{RTE}^{(1)}$ | 0.870 | 1.000 | 0.910 |
| | $P(RTE>0.75)$ | 0.994 | 1.000 | 0.998 |
| | Error $(\overline{\varepsilon})^{(2)}$ | 0.840 | 0.000 | 0.730 |
| **12** | $\overline{RTE}^{(1)}$ | 0.570 | 0.710 | 0.810 |
| | $P(RTE>0.75)$ | 0.006 | 0.034 | 0.778 |
| | Error $(\overline{\varepsilon})^{(2)}$ | 1.150 | 0.440 | 1.160 |
| **13** | $\overline{RTE}^{(1)}$ | 0.940 | 0.900 | 0.910 |
| | $P(RTE>0.75)$ | 1.000 | 1.000 | 1.000 |
| | Error $(\overline{\varepsilon})^{(2)}$ | 0.390 | 0.390 | 0.740 |

[1] The maximum RTE on average is 1, and the minimum is 0.

[2] $\overline{\varepsilon} = (\varepsilon \times 100)/\overline{RTE}$, the error on average ($\overline{\varepsilon}$) is a percentage (%) of $\overline{RTE}$. To achieve good accuracy, $\overline{\varepsilon} < 2.5\%$.

When $\overline{RTE}$ was analysed, only areas 3 and 13 appeared as the best for the two main types of care and were very good in the third type (they can be considered global benchmarks in the ecosystem). Any area can be considered worst for the three types of care, as there was no global target-for-improvement area.

For acute hospital care (S1), areas 4 and 9 showed the lowest RTE errors on average ($\overline{\varepsilon}$) (simulation results are more accurate), while areas 7, 10 and 12 had the highest ones (simulation results are more dispersed) (Table 3). For core health day care (S2), areas 1, 5, 6 and 11 had the lowest $\overline{\varepsilon}$, while area 3 achieved the highest value. Finally, for outpatient care (S3), areas 3, 4, 5 and 7 were the best (lowest $\overline{\varepsilon}$), whereas areas 2, 8, 9 and 12 were the worst. None of the $\overline{\varepsilon} \geq 2.5\%$; therefore, from a statistical perspective, the simulation results can be considered appropriate.

Areas 1, 2, 3, 5, 9 and 13 showed the best probability of having an RTE greater than 0.75, $P$($RTE$>0.75), in acute hospital care (S1), while areas 7 and 12 had the worst probability (Table 3). For core health day care (S2), areas 3, 4, 11 and 13 reached the highest $P$($RTE$>0.75), while areas 1, 2, 5, 6, 7, 8, 9 and 10 showed the lowest $P$ (zero). Finally, for outpatient care (S3), areas 4, 5, 7 and 13 were the best, whereas areas 1 and 2 were the worst.

## The stability of the MH system in Gipuzkoa

The Mental Health System of Gipuzkoa is likely unstable. For acute hospital care (S1), areas 1, 6, 8 and 9 were the most stable, while areas 7 and 12 were the most unstable (Table 4). For core health day care (S2), areas 2, 9, and 11 could be considered the most stable areas. In contrast, the area 5 was likely unstable. Finally, areas 3 and 4 were likely stable for outpatient care (S3), but areas 6 and 8 were very unstable.

## The entropy of the MH system in Gipuzkoa

For acute hospital care (S1), areas 3, 7, 8 and 9 showed the least entropy, while areas 2, 4, 5, 10, 11 and 12 had the most entropy (Table 5). Core health day care (S2) presented the lowest entropy values in areas 1, 2, 6, 7, 9, 10 and 11, while areas 3, 4 and 13 showed the highest

**Table 4. Stability results.** White indicates the best scores, and dark grey indicates the worst scores. Darker shading means lower stability.

| Stability [1] | Acute hospital care scenario | Core health day care scenario | Outpatient care scenario |
|---|---|---|---|
| Area 1 | 68.93 | 68.49 | 60.97 |
| Area 2 | 59.12 | 75.95 | 49.95 |
| Area 3 | 61.06 | 69.04 | 64.27 |
| Area 4 | 56.85 | 70.27 | 69.97 |
| Area 5 | 58.91 | 54.85 | 63.79 |
| Area 6 | 64.8 | 71.28 | 30.74 |
| Area 7 | 42.33 | 69.05 | 62.13 |
| Area 8 | 63.35 | 64.94 | 32.94 |
| Area 9 | 72.04 | 73.81 | 52.61 |
| Area 10 | 52.83 | 69.05 | 60.66 |
| Area 11 | 56.71 | 96.52 | 56.17 |
| Area 12 | 42.97 | 64.38 | 54.33 |
| Area 13 | 58.36 | 70.49 | 60.11 |

[1] The stability is assessed as a percentage (%), the maximum stability is 100% (the ecosystem is completely stable), and the minimum is 0% (the ecosystem is completely unstable).

**Table 5. Entropy scores.** White indicates the lowest values, and dark grey indicates the highest values. Darker shading means higher entropy.

| Entropy [1] | Acute hospital care scenario | Core health day care scenario | Outpatient care scenario |
|---|---|---|---|
| Area 1 | 44.8 | 5.02 | 20.08 |
| Area 2 | 55.09 | 2.65 | 48.26 |
| Area 3 | 41.30 | 44.27 | 46.61 |
| Area 4 | 54.34 | 37.40 | 41.3 |
| Area 5 | 53.39 | 22.36 | 38.85 |
| Area 6 | 46.12 | 1.81 | 80.13 |
| Area 7 | 38.99 | 0.00 | 49.36 |
| Area 8 | 40.85 | 11.07 | 77.46 |
| Area 9 | 33.09 | 0.00 | 53.05 |
| Area 10 | 57.94 | 0.00 | 46.01 |
| Area 11 | 54.97 | 8.21 | 55.07 |
| Area 12 | 53.21 | 25.63 | 57.95 |
| Area 13 | 49.28 | 35.21 | 49.19 |

[1] Shannon´s entropy is assessed as a percentage (%) of its mathematical maximum; the maximum entropy is 100% (the ecosystem exhibits complete heterogeneous management), and the minimum is 0% (the ecosystem exhibits complete homogeneous management).

entropy values. Finally, for outpatient care (S3), areas 1 and 5 had the lowest entropy values, whereas areas 6 and 8 had the highest entropy values.

## The performance of the MH system: Summarizing all the indicators to identify benchmarks and targets for improvement in order to advance tailor-made potential interventions

For acute hospital care (S1), area 9 showed the best global performance (it is the benchmark), with a high $\overline{RTE}$ with a low error and a high $P(RTE>0.75)$ and stability, and relatively low data variability (Tables 1 to 3). In this scenario, area 1, on a second level, and areas 3 and 13, on a third level, can also be considered potential benchmarks. In the former, this area has a high $\overline{RTE}$ ($\overline{\varepsilon}$ was not very low), $P(RTE>0.75)$ and stability. In the latter, both areas have a high $\overline{RTE}$ (but the $\overline{\varepsilon}$ was not very low) and $P(RTE>0.75)$, but they are likely unstable (Tables 1 to 3).

Area 11 is the benchmark for core health day care (S2) for the same reasons as area 9 was the benchmark in the previous study (Tables 1 to 3). Area 13 can also be considered a potential benchmark for its high $\overline{RTE}$ ($\overline{\varepsilon}$ was not very low) and $P(RTE>0.75)$, but its data variability is not very good.

For outpatient care (S3), area 4 is the global benchmark because it has the same characteristics as areas 9 (acute hospital care) and 11 (core health day care), but its data variability is likely high (Tables 1 to 3). On a second level, area 3 can be considered a potential benchmark because it has high $\overline{RTE}$ ($\overline{\varepsilon}$ was low) and stability, its $P(RTE>0.75)$ score is intermediate, and its data variability is likely high. On the third level, area 5 can also be a potential benchmark because of its high $\overline{RTE}$ ($\overline{\varepsilon}$ was low) and $P(RTE>0.75)$, but it is likely unstable, and its data variability is high.

MH provision (availability, beds and professional rates) in area 7 (target-for-improvement area) in acute hospital care can be considered scarce in comparison to the benchmarks, except for area 1, which has a very similar profile (Fig 1A). The potential increase in these variables seems to be relevant, reaching 100% when area 7 is compared to area 9. When compared to

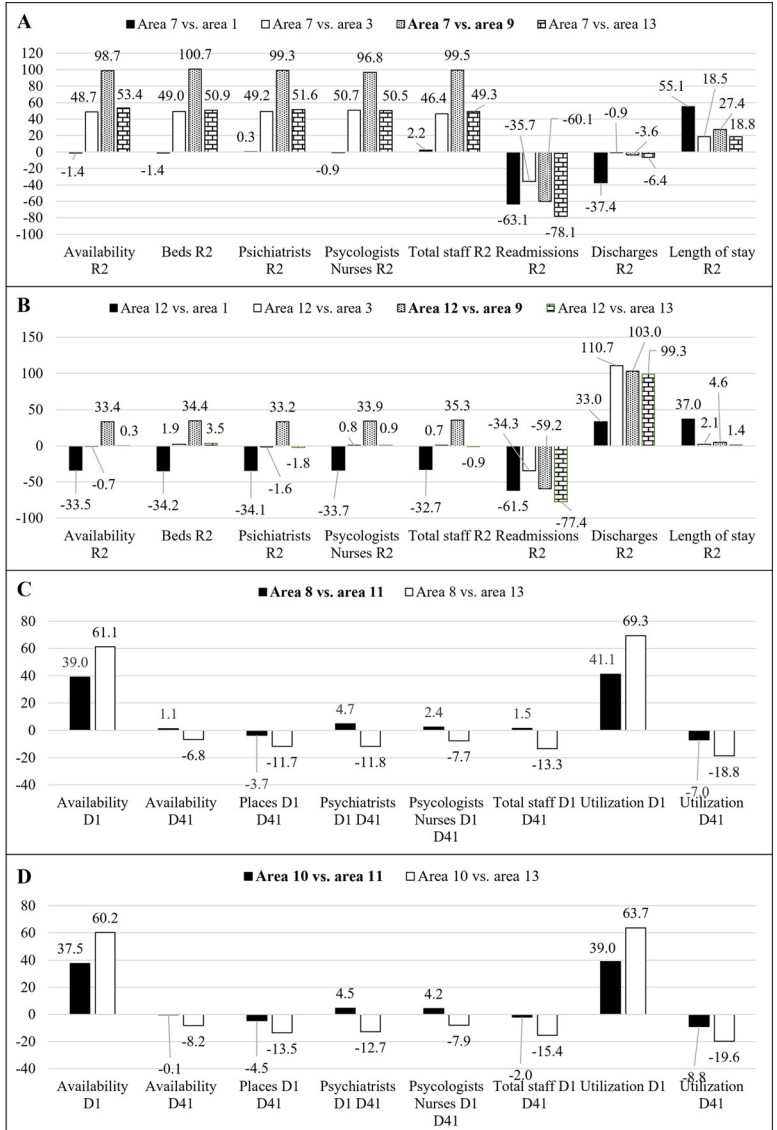

**Fig 1.** A) Acute hospital care (R2 code, DESDE-LTC) scenario, potential improvements (in percentage, %) for area 7. B) Acute hospital care (R2 code, DESDE-LTC) scenario, potential improvements (in percentage, %) for area 12. C) Core health day care (D1 and D4.1 codes, DESDE-LTC) scenario, potential improvements (in percentage, %) for area 8. D) Core health day care (D1 and D4.1 codes, DESDE-LTC) scenario, potential improvements (in percentage, %) for area 10. Note: The target area for improvement has been compared to the selected benchmark (global benchmark for the scenario is bolded).

area 1, area 7's readmission and discharge scores are very high combined with a limited length of stay; the former should be reduced, and the latter should be increased to improve area 7's performance.

In Gipuzkoa, acute hospital care is located in areas 2, 6 and 9. Data from these facilities were distributed proportionally in the remainder of the areas under study. Considering this procedure, when a provision increase (rate) is needed, decision-makers can (i) expand the service (expensive and opposite to community-based care), (ii) rebalance target populations or (iii) both. These interventions must be carefully studied because they can modify the whole ecosystem structure as well as its administration and consequently its global performance.

The acute hospital care profile of area 12 (target-for-improvement) is more complicated (Fig 1B). Depending on the selected benchmark, provision rates can decrease (area 1), increase (area 9) or remain constant (areas 3 and 13), and there are different equilibria. Nevertheless, the key for increasing the performance of area is to reach a specific value for readmissions (decreasing), discharges (increasing) and length of stay (increasing) scores. For example, if acute care provision must remain constant, the readmission score should decrease and the number of discharges should increase proportionally (i.e., if the former decreases less, the latter should increase more, Fig 1B).

For core health day care, four areas can be considered target-for-improvement areas: 1, 7, 8 and 10. However, there are only two structural situations that could define potential interventions. The first one is defined by areas 8 and 10. In this case, the availability and utilization scores of acute day services (D1 code, DESDE-LTC) should increase significantly (Fig 1C and 1D). In these catchment areas, as mentioned before, core health day care facilities are integrated in their respective MH centres (which provide outpatient care). This unique structure, which increases care coordination, tends to overshadow both availability and utilization scores in areas 8 and 10 (Fig 1C and 1D).

The second potential intervention is characterized by the profiles of areas 1 and 7. In this situation, the availability and utilization scores are less relevant. The availability rate of non-acute day services (D4.1 code, DESDE-LTC), rate of professionals (integrating D1 and D4.1 services) and utilization rate should be increased (Fig 2A and 2B). These increases could be more moderate if the availability and utilization scores for acute day services (D1) also increase slightly, balancing the global core health day care provision.

For outpatient care, areas 1 and 2 are identified as the target for improvement areas. In both cases, a significant increase in the availability rate should be considered to improve performance. If the intervention greatly increases the availability rate, then the corresponding psychologist and nurse rates should also be increased. In contrast, if the prior increase is moderate, the availability of the mentioned health professionals could be considered overestimated and should be reduced (Fig 2C and 2D). In all areas, an increase in the incidence is expected to improve performance. This phenomenon could require a target population reorganization (increase the number of persons in each target-for-improvement area) that should involve a restructuration of the MH areas of Gipuzkoa.

## Discussion

To the best of our knowledge, this is the first study that identifies benchmark and target-for-improvement areas in a real-world MH ecosystem. It is worth highlighting that the Gipuzkoa management strategy follows the balanced care model [3], trying to achieve an equilibrium among hospital and community care. In this respect, 90% of the activity developed by the Mental Health System of Gipuzkoa is non-acute, non-mobile outpatient care, which indicates a high tendency toward community care [36]. At the same time, the current management strategy is focused on increasing acute core health day care by developing a new day hospital as an alternative to hospitalization and continuity of care.

From a policy perspective, this research facilitates evidence-informed decision-making to improve the continuity of care after hospitalization. Our findings demonstrate the feasibility of a DSS for helping design interventions and policies to improve MH care to increase integration and balance [37].

The potential improvement strategies proposed are aligned with the balanced MH care model [38] in each scenario (acute inpatient care, acute and non-acute core health day care and non-acute and non-mobile outpatient care). In this study, potential interventions are

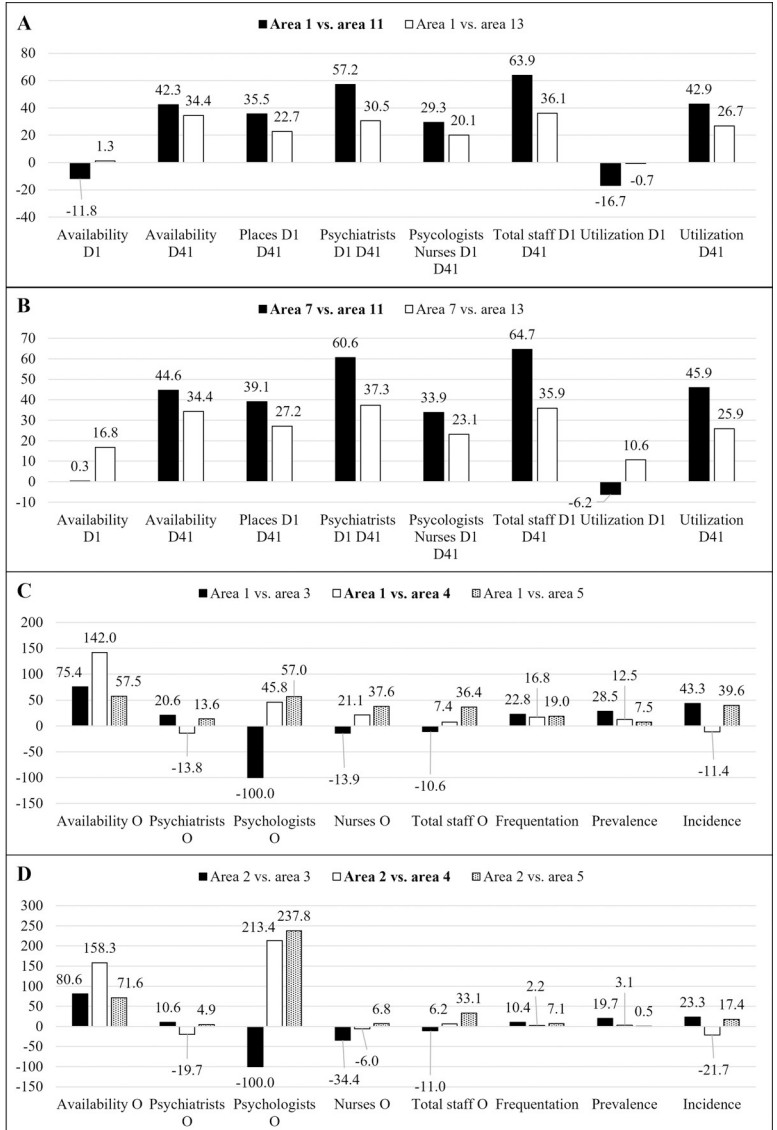

**Fig 2.** A) Core health day care (D1 and D4.1 codes, DESDE-LTC) scenario, potential improvements (in percentage, %) for area 1. B) Core health day care (D1 and D4.1 codes, DESDE-LTC) scenario, potential improvements (in percentage, %) for area 7. C) Outpatient (O8 to O10 codes, DESDE-LTC) care scenario, potential improvements (in percentage, %) for area 1. D) Outpatient (O8 to O10 codes, DESDE-LTC) care scenario, potential improvements (in percentage, %) for area 2. Note: The target area for improvement has been compared to the selected benchmark (global benchmark for the scenario is bolded).

based mainly on resource allocation, although previous research on RTE assessment of MH services has identified improvement strategies that involve reducing beds, staff and supply expenses [39], budget allocated for administrative services and full-time equivalent personnel [40] or operating costs [41].

Comparing the characteristics of benchmark and target-for-improvement areas, structural differences can be used to guide decision-making because decision-makers can assess structural (i.e., hide more professionals), administrative (i.e., modify the target population for a catchment area) and procedural (i.e., change admission processes) interventions for improving MH care provision. These proposals must be modulated over time because they are infeasible

to perform simultaneously. Thus, it is necessary to assess not only individual performances but also their potential impact on the other parts of the MH care ecosystem. In this process, expert knowledge is essential because no mathematical or operational method can interpret the nuances that characterize any MH ecosystem under a specific care paradigm. By integrating expert knowledge, decision-making becomes deductive, based on data, and inductive, based on informed evidence [42]. This fact is especially relevant when raw data have been transformed to analyse specific observation sets [21].

In this study, 13 catchment areas were selected to assess the MH ecosystem status and to propose potential interventions to improve it. All these areas have their own MH centre; thus, outpatient care data have been directly assigned. However, only a few areas have acute hospital (three areas) and/or core health day care facilities. To make the analysis possible, data from these areas were distributed proportionally. Therefore, the dataset corresponding to these MH types is a realistic estimation verified by the decision-makers of the Mental Health Network of Gipuzkoa.

The use of a DSS allowed ecosystem performance to be assessed and potential evidence-informed interventions to be identified for target-for-improvement areas [21,24,29].

The profiles of the target-for-improvement areas in acute inpatient care (R2, DESDE-LTC code) are very different. Area 7 could require a significant increase in the number of beds and/or in the rates (per 100,000 adult inhabitants) of psychiatrists and psychologist-nurses. Obviously, a structural intervention such as this must be modulated over time, expecting both a substantial decrease in the readmission rate and a moderate increase in the length of stay. In area 12, the input structure (availability, bed, psychiatrist, and psychologist-nurse rates) can be considered likely appropriate, but a significant decrease in its readmission rate and a very high increase in discharges could be needed. Its profile indicates that it is probably unnecessary to modify the structure of the MH care provided in this area, but a modification of the procedures could be considered. Considering that acute hospital data have been calculated by distributing raw data from areas 2, 6 and 9, all previous interventions must be transferred to these facilities. Moreover, from a mathematical viewpoint, a rebalance of the target population for each catchment area (especially in the target-for-improvement areas) can be the easiest way to improve ecosystem performance because it would modify the corresponding input rates. Certainly, modifying the structure of the acute hospital facilities (there is only one facility in Gipuzkoa) can also rebalance the current situation, but this intervention is more labour-intensive for the organization.

The DSS identified four target-for-improvement areas in core health day care provision. In the first area (area 8), the availability rate should be significantly increased. To realize such an improvement, there are two options: enable new day services (probably only D1 because an increase in its utilization score is also considered appropriate) and/or decrease the target population (rebalancing it between neighbouring areas). The first option is a structural intervention, probably expensive and unreasonable but, from a theoretical viewpoint, can have an immediate positive impact on MH care provision, especially if a D4.1 facility is transformed into a D1 facility. Second, administrative intervention is less effort-intensive for the organization, but its potential positive effect will be delayed (users have strong inertia, so it can be difficult to convince them to change and visit other services). The second area (area 10) showed a very similar profile; therefore, all the considerations made for area 8 are applicable here. Areas 8 and 10 have integrated core health day care facilities into their MH centres. This structure is especially relevant to understand the proposed interventions because, for this reason, their availability and utilization scores were probably underestimated.

In areas 1 and 7, the DSS identified a problem in the provision of rehabilitation core health day care (D4.1), which should be reinforced in both areas. Availability and place rates

(capacity) as well as psychiatrist, psychologist, and nurse rates in D4.1 service should be significantly increased in these areas. Except for the availability rate, where the comments made for areas 8 and 10 are valid, the other potential interventions are structural (the number of MH professionals should be increased). Again, efforts made by decision-makers should be rewarded by an increase in the utilization of the D4.1 provision. Here, it can also be helpful to transform a D1 service (or a section) in a D4.1 facility. In this intervention, a decrease in the utilization of D1 services and an increase in the corresponding utilization in non-acute core health day care services are expected (both with a potential positive impact).

In non-acute and non-mobile outpatient care services, indicators highlighted two target-for-improvement areas (1 and 2), both with similar profiles. Here, the most relevant variables are availability and psychologist rates. For the first area, the methodology recommends a significant increase. This outcome can be achieved by opening new inpatient care services and/or reducing (rebalancing) the target population. The recommendation focused on reinforcing inpatient care services does not meet the objectives established by current MH care models, which promote the development of community-based care services. Notably, MH residential care services can reach high levels of RTE because they are highly structured, and the management strategies are strongly homogenous. For this reason, the DSS recommends reinforcing inpatient care. The effects of these interventions were described in the previous paragraph.

According to data values and depending on the benchmark area, the psychologist rate showed contradictory behaviour. In area 3, outpatient care is based mainly on psychiatrists because for areas 1 and 2, the comparison recommends a significant decrease in the psychologist rate, but it also expects a small increase in the frequency. This intervention cannot be considered appropriate for areas 1 and 2 because their structures are more similar to benchmark areas 4 and 5. Carrying out the respective comparisons, a significant increase in their psychologist rates is highlighted, thus predicting a relevant increase in their frequentation and prevalence. Depending on the prior increase, the expected incidence should increase (higher psychologist rate variation) or decrease (lower psychologist rate variation).

The performance of the Mental Health Network of Gipuzkoa in each scenario could be increased by implementing the recommended organizational interventions. The analytical process followed in this study can be used as a new tool to reach a more integrated and balanced MH care system considering the population needs and evidence from the data and DSS.

However, the present study had some limitations. The analyses only included 25 variables that offered a relatively precise view of the real ecosystem situation, but it would be necessary to include new variables, mainly outcomes, for a more precise benchmarking analysis. The results offer an initial view of the situation from three single perspectives (scenarios), but new perspectives must be designed to analyse the real balance of care in each catchment area as well as in the Gipuzkoa MH system. Considering that some of the proposed interventions may be appropriate in some scenarios but inadequate in others (by modifying the status of the MH care balance), it is necessary to refine the analytical tools to improve understanding of the interventions' potential impact on the whole MH ecosystem.

The original dataset given by the Mental Health system of Gipuzkoa is completely reliable, but the main bias source is population. In order to calculate the corresponding availability and resource's use rates the most reliable source for population is the official census, that is relatively outdated. In order to deal with this problem, uncertainty, a Monte-Carlo simulation engine has been developed to manage statistical distributions rather than raw data. On the other hand, data on MH systems are dynamic and change frequently the analysis showed in this paper corresponds to a specific transversal cut in the time but the DSS can analyse new datasets very easily. The Monte-Carlo simulation engine offers a sensitivity analysis to take

relative small data changes into control by multiplying the number of observation artificially [21].

## Conclusions

This research has contributed to the development of evidence-informed policymaking by including scientific findings to deal with the inner uncertainty of the environment as well as context information (expert knowledge for interpreting data under a theoretical paradigm). The formal integration of evidence (data) and expert knowledge is the most powerful strategy to increase expert knowledge, and expert knowledge is the unique tool to deal with uncertainty in a real and complex ecosystem. The proposed DSS has allowed firstly to assess both the individual (decision units) and global performance of the MH system of Gipuzkoa and secondly to identify new potential improvement strategies prior its potential implementation in the real environment, this process decreases operational risks. This article can be considered a guide for informing experts, managers and policy makers on how real MH systems are really working (performance) and how they could be improved to provide better MH care.

Gipuzkoa follows the balanced care model, with a strong orientation to provide non-acute and non-mobile outpatient care, which is the main activity developed by its mental health system. The analytical procedure designed in this study focused on three specific scenarios that represent the main types of MH care: inpatient, core health day and outpatient care. From these three perspectives, no catchment area must be organizationally improved in its three main types of care. Area 1 has a strong residential care structure, but its day and outpatient facilities may be improved. Area 2 has a weak outpatient care sub-ecosystem, but from residential and day care viewpoints, it is strong. Area 7 should improve residential care (rates), and in area 10, day care is the target for improvement. Finally, in area 12, residential care is the weakness.

No area can be considered the global benchmark for Gipuzkoa. Area 13 can be considered a reference for both residential and day care and area 3, for residential and outpatient care. However, the best areas for the three main MH care types, acute hospital care, day care, and outpatient care, are areas 9, 11 and 4, respectively.

Considering the current situation of MH care, improved planning and management methods are required to meet users' needs. Despite efforts since the 1950s with respect to the deinstitutionalization process for improving MH systems, there remains a need for new policies that facilitate the continuity of community care for users who have been discharged from hospitals. DSSs allow the assessment of the MH ecosystem performance as well as the identification of benchmark and target-for-improvement catchment areas. The potential improvement strategies identified for each area were designed according to the balanced MH care model specific to each scenario (inpatient, core health day care and outpatient care).

In this analysis, we did not follow a "whole system approach" but instead adopted the perspective of the planning agency of Gipuzkoa. Therefore, we did not include in the DSS all service provisions for MH care beyond the main types of care. Psychologists and nurses were grouped in acute inpatient care, so better granularity is recommended for a better intervention design.

## Acknowledgments

We would like to thank Álvaro Iruin and Andrea Gabilondo (Mental Health Network of Gipuzkoa) as well as Carlos Pereira and José Juan Uriarte (Mental Health Network of Bizkaia) for sharing your knowledge and improving this research.

## Author Contributions

**Conceptualization:** Carlos R. García-Alonso, Nerea Almeda, Álvaro Iruin-Sanz.

**Data curation:** Carlos R. García-Alonso, Nerea Almeda, José A. Salinas-Pérez, Mencía R. Gutiérrez-Colosía.

**Formal analysis:** Carlos R. García-Alonso, Nerea Almeda.

**Funding acquisition:** Carlos R. García-Alonso.

**Investigation:** Carlos R. García-Alonso, Nerea Almeda.

**Methodology:** Carlos R. García-Alonso, Nerea Almeda.

**Resources:** Nerea Almeda.

**Software:** Nerea Almeda.

**Supervision:** Carlos R. García-Alonso, Nerea Almeda, Álvaro Iruin-Sanz, Luis Salvador-Carulla.

**Validation:** Nerea Almeda, Álvaro Iruin-Sanz, Luis Salvador-Carulla.

**Visualization:** Nerea Almeda.

**Writing – original draft:** Carlos R. García-Alonso, Nerea Almeda.

**Writing – review & editing:** Carlos R. García-Alonso, Nerea Almeda, José A. Salinas-Pérez, Mencía R. Gutiérrez-Colosía, Álvaro Iruin-Sanz, Luis Salvador-Carulla.

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
