## [Decision Letter · Decision Letter 0]

5 Jan 2022

PONE-D-21-30492

Use of a decision support system for benchmarking analysis and organizational improvement of regional mental health care: The case of Gipuzkoa (Basque Country, Spain)

PLOS ONE

Dear Dr. Almeda,

Thank you for submitting your manuscript to PLOS ONE. After careful consideration, we feel that it has merit but does not fully meet PLOS ONE’s publication criteria as it currently stands. Therefore, we invite you to submit a revised version of the manuscript that addresses the points raised during the review process.

We look forward to receiving your revised manuscript.

Kind regards,

Majid Soleimani-damaneh

Academic Editor

PLOS ONE

https://journals.plos.org/plosone/s/file?id=ba62/PLOSOne_formatting_sample_title_authors_affiliations.pdf"

“This study was partially funded by a Carlos III Health Institute grant (PI18/01521) and the Regional Government of Andalusia (PY18-RE-0022) with European Union FEDER teams.

The funding agreement ensured the authors’ independence in designing the study, interpreting the data, writing, and publishing the report.”

“We would like to thank the Mental Health Network of Gipuzkoa, especially Álvaro Iruin and Andrea Gabilondo, for providing data and supporting this study. We also acknowledge the Mental Health Network of Bizkaia, especially Carlos Pereira and José Juan Uriarte, for supporting this research.”

“This study was partially funded by a Carlos III Health Institute grant (PI18/01521) and the Regional Government of Andalusia (PY18-RE-0022) with European Union FEDER teams.

The funding agreement ensured the authors’ independence in designing the study, interpreting the data, writing, and publishing the report.”

Reviewers' comments:

Reviewer's Responses to Questions

**Comments to the Author**

1. Is the manuscript technically sound, and do the data support the conclusions?

Reviewer #1: Yes

Reviewer #2: Yes

2. Has the statistical analysis been performed appropriately and rigorously? 

Reviewer #1: Yes

Reviewer #2: I Don't Know

3. Have the authors made all data underlying the findings in their manuscript fully available?

Reviewer #1: No

Reviewer #2: Yes

4. Is the manuscript presented in an intelligible fashion and written in standard English?

Reviewer #1: Yes

Reviewer #2: Yes

5. Review Comments to the Author

Reviewer #1: The manuscript is not well organized. The explanation of the manuscript contribution should be stronger, especially for a wider range of readers. Think about how readers' access to the statistics and information you used. Are decision support systems suitable for all decisions? I think you should make a logical argument (not just referring to other sources) as to why the decision support system is useful for your case.

Reviewer #2: This is an interesting manuscript offering the readers a good and thorough analysis to develop an analytical process for (i) assessing the performance of Mental Health in Gipuzkoa (Basque Country, Spain) and identifying benchmark and target-for-improvement catchment areas.

According to the authors, the main conclusion of the study is that no catchment area could be reference in its three main types of care.

The article has interesting content, although I have highlighted some issues below:

I suggest indicate the study’s design in the title or the abstract.

The manuscript can still be improved by including a more in-depth discussion (including direction and magnitude ) of potential bias, imprecision or confounders.

6. PLOS authors have the option to publish the peer review history of their article (what does this mean?). If published, this will include your full peer review and any attached files.

Reviewer #1: No

Reviewer #2: No

---

## [Author Response · Author response to Decision Letter 0]

28 Jan 2022

Response to reviewers

Journal requirements

https://journals.plos.org/plosone/s/file?id=ba62/PLOSOne_formatting_sample_title_authors_affiliations.pdf"

Many thanks for your assistance, we have ensured that the manuscript meets PLOS ONE’s style.

“This study was partially funded by a Carlos III Health Institute grant (PI18/01521) and the Regional Government of Andalusia (PY18-RE-0022) with European Union FEDER teams. The funding agreement ensured the authors’ independence in designing the study, interpreting the data, writing, and publishing the report.”

Thank you very much for your assistance. We have corrected the mistake and follow the instructions according to Plos One journal. We have also included the updated Funding Statement in the cover letter:

This study was funded by the Instituto de Salud Carlos III (grant number: PI18/01521; author who received the award: CGA) https://www.isciii.es/Paginas/Inicio.aspx and Junta de Andalucía con Fondos FEDER (Unión Europea) (grant number: Y18-RE-0022; author who received the award: CGA) https://www.juntadeandalucia.es/economiaconocimientoempresasyuniversidad/fondoseuropeosenandalucia/feder. There was no additional external funding received for this study. The funders had no role in study design, data collection and analysis, decision to publish, or preparation of the manuscript.

3. Thank you for stating the following in the Acknowledgments Section of your manuscript: “We would like to thank the Mental Health Network of Gipuzkoa, especially Álvaro Iruin and Andrea Gabilondo, for providing data and supporting this study. We also acknowledge the Mental Health Network of Bizkaia, especially Carlos Pereira and José Juan Uriarte, for supporting this research.” We note that you have provided funding information that is not currently declared in your Funding Statement. However, funding information should not appear in the Acknowledgments section or other areas of your manuscript. We will only publish funding information present in the Funding Statement section of the online submission form. Please remove any funding-related text from the manuscript and let us know how you would like to update your Funding Statement. Currently, your Funding Statement reads as follows: “This study was partially funded by a Carlos III Health Institute grant (PI18/01521) and the Regional Government of Andalusia (PY18-RE-0022) with European Union FEDER teams. The funding agreement ensured the authors’ independence in designing the study, interpreting the data, writing, and publishing the report.” Please include your amended statements within your cover letter; we will change the online submission form on your behalf.

We are sorry for the mistakes; we have updated the Funding Statement in the cover letter as follow and we would like this paragraph to appear in the Funding Statement:

This study was funded by the Instituto de Salud Carlos III (grant number: PI18/01521; author who received the award: CRGA) https://www.isciii.es/Paginas/Inicio.aspx and Junta de Andalucía con Fondos FEDER (Unión Europea) (grant number: Y18-RE-0022; author who received the award: CRGA) https://www.juntadeandalucia.es/economiaconocimientoempresasyuniversidad/fondoseuropeosenandalucia/feder . There was no additional external funding received for this study. The funders had no role in study design, data collection and analysis, decision to publish, or preparation of the manuscript.

We have also updated the Acknowledgement Section (in yellow in the manuscript):

We would like to thank Álvaro Iruin and Andrea Gabilondo (Mental Health Network of Gipuzkoa) as well as Carlos Pereira and José Juan Uriarte (Mental Health Network of Bizkaia) for sharing your knowledge and improving this research.

Reviewers' comments

Reviewer #1: The manuscript is not well organized.

Thank you very much for your suggestion. We have modified the “aims” paragraph in the introduction adding the following explanation (in yellow in the manuscript).

According to this, the paper is organised in the following sections: (i) Methods, to firstly describe the real system structure and the available dataset (variables) and secondly to set the basic indicator definitions and briefly analyse the usability of the DSS in supporting the analysis of real interventions and policies, (ii) Results, analysing the MH system performance (relative technical efficiency, stability and entropy) and identifying potential interventions on “target for improvement areas” considering the structure of “benchmarking” ones and, finally, (iii) this paper ends with the discussion and conclusion sections.

We have also re-structured the Method section by including a new section at the end “Potential usability of the DSS in real complex systems” where we have included the following paragraph:

DSSs systems are considered appropriate tools for guiding operational (basically resource allocation and use) evidence-informed planning and management. DSSs allow decision makers and policy makers to increase their knowledge about how the organization of the ecosystem (divided in comparable organizational units) manages scarce resources in order to produce mensurable and positive results, in the end: the ecosystem performance. Commonly, decisions and policymaking are based on the manager’s experience, external opinions, historical facts, etc., and they need new evidence-based tools for increasing their knowledge when organizational interventions or policies must be assessed in advance. The DSS proposed in this paper allows them to have an objective assessment about the ecosystem performance including relative technical efficiency, stability and entropy, everything from different points of view. In addition, this computer-based tool can identify new improvement strategies once the proposed changes had been potentially taken in the real ecosystem, aiming at improving MH care provision.

Also, in the Method section we have modify the second subsection to “Indicator definitions”.

Finally, the “Results” section has been revised by adding four new subsections trying to clarify its content, on one hand to emphasized relative technical efficiency, stability and entropy results and, on the other, to identify both “benchmarking” and “target for improvement” areas in order to propose potential interventions in order to reach better mental health care.

The explanation of the manuscript contribution should be stronger, especially for a wider range of readers. 

Many thanks, we have carried out this change in the Conclusions section (highlighted in yellow in the manuscript):

This research has contributed to the development of evidence-informed policymaking by including scientific findings to deal with the inner uncertainty of the environment as well as context information (expert knowledge for interpreting data under a theoretical paradigm). The formal integration of evidence (data) and expert knowledge is the most powerful strategy to increase expert knowledge, and expert knowledge is the unique tool to deal with uncertainty in a real and complex ecosystem. The proposed DSS has allowed firstly to assess both the individual (decision units) and global performance of the MH system of Gipuzkoa and secondly to identify new potential improvement strategies prior its potential implementation in the real environment, this process decreases operational risks. This article can be considered a guide for informing experts, managers and policy makers on how real MH systems are really working (performance) and how they could be improved to provide better MH care.

Think about how readers' access to the statistics and information you used. 

Many thanks for your suggestion. We have included this modification in the text (Methods> Study area and data): 

Data set is available at the Dryad digital repository (https://datadryad.org/stash/share/cFD5jA5F3odtAhZqUARiPBTo_yj8lKvQ9GWAGrauPnE) and processed datasets that feed the DSS are available upon request. 

Are decision support systems suitable for all decisions? I think you should make a logical argument (not just referring to other sources) as to why the decision support system is useful for your case.

Many thanks, we have carried out this suggestion (Methods > Potential usability of the DSS in real complex systems, in yellow):

DSSs systems are considered appropriate tools for guiding operational (basically resource allocation and use) evidence-informed planning and management. DSSs allow decision makers and policy makers to increase their knowledge about how the organization of the ecosystem (divided in comparable organizational units) manages scarce resources in order to produce mensurable and positive results, in the end: the ecosystem performance. Commonly, decisions and policymaking are based on the manager’s experience, external opinions, historical facts, etc., and they need new evidence-based tools for increasing their knowledge when organizational interventions or policies have to be assessed in advance. The DSS proposed in this paper allows them to have an objective assessment about the ecosystem performance including relative technical efficiency, stability and entropy, everything from different points of view. In addition, this computer-based tool can identify new improvement strategies once the proposed changes had been potentially taken in the real ecosystem, aiming at improving MH care provision.

Reviewer #2: This is an interesting manuscript offering the readers a good and thorough analysis to develop an analytical process for (i) assessing the performance of Mental Health in Gipuzkoa (Basque Country, Spain) and identifying benchmark and target-for-improvement catchment areas.

According to the authors, the main conclusion of the study is that no catchment area could be reference in its three main types of care.

The article has interesting content, although I have highlighted some issues below:

I suggest indicate the study’s design in the title or the abstract.

Many thanks for your suggestion, we have modified the title as follow:

Use of a decision support system for benchmarking analysis and organizational improvement of regional mental health care: efficiency, stability and entropy assessment of the Mental Health ecosystem of Gipuzkoa (Basque Country, Spain)

The manuscript can still be improved by including a more in-depth discussion (including direction and magnitude) of potential bias, imprecision or confounders.

Many thanks for this suggestion, we have carried out the proposed change (in yellow in the manuscript). 

The original dataset given by the Mental Health system of Gipuzkoa is completely reliable, but the main bias source is population. In order to calculate the corresponding availability and resource’s use rates the most reliable source for population is the official census, that is relatively outdated. In order to deal with this problem, uncertainty, a Monte-Carlo simulation engine has been developed to manage statistical distributions rather than raw data. On the other hand, data on MH systems are dynamic and change frequently the analysis showed in this paper corresponds to a specific transversal cut in the time but the DSS can analyse new datasets very easily. The Monte-Carlo simulation engine offers a sensitivity analysis to take relative small data changes into control by multiplying the number of observation artificially (21).

Thank you very much for your assistance. We have upload the figure files to the Preflight Analysis and Conversion Engine (PACE) digital diagnostic tool and they meet PLOS requirements.

---

## [Decision Letter · Decision Letter 1]

7 Mar 2022

Use of a decision support system for benchmarking analysis and organizational improvement of regional mental health care: efficiency, stability and entropy assessment of the Mental Health ecosystem of Gipuzkoa (Basque Country, Spain)

PONE-D-21-30492R1

Dear Dr. Almeda,

We’re pleased to inform you that your manuscript has been judged scientifically suitable for publication and will be formally accepted for publication once it meets all outstanding technical requirements.

Kind regards,

Majid Soleimani-damaneh

Academic Editor

PLOS ONE

Additional Editor Comments (optional):

Reviewers' comments:

Reviewer's Responses to Questions

**Comments to the Author**

1. If the authors have adequately addressed your comments raised in a previous round of review and you feel that this manuscript is now acceptable for publication, you may indicate that here to bypass the “Comments to the Author” section, enter your conflict of interest statement in the “Confidential to Editor” section, and submit your "Accept" recommendation.

Reviewer #1: All comments have been addressed

Reviewer #2: All comments have been addressed

2. Is the manuscript technically sound, and do the data support the conclusions?

Reviewer #1: Yes

Reviewer #2: Yes

3. Has the statistical analysis been performed appropriately and rigorously? 

Reviewer #1: I Don't Know

Reviewer #2: Yes

4. Have the authors made all data underlying the findings in their manuscript fully available?

Reviewer #1: No

Reviewer #2: Yes

5. Is the manuscript presented in an intelligible fashion and written in standard English?

Reviewer #1: Yes

Reviewer #2: Yes

6. Review Comments to the Author

Reviewer #1: The use of decision support systems in such a case requires more reasoning and mere citation is not enough. In addition, more reasoning is needed to determine the operational value-added of the output. Unfortunately, when we look at the results, we see that the results are not something that is unknown without doing this research, while the topic you have chosen potentially has much deeper and newer consequences.

Reviewer #2: (No Response)

7. PLOS authors have the option to publish the peer review history of their article (what does this mean?). If published, this will include your full peer review and any attached files.

Reviewer #1: No

Reviewer #2: No

---

## [Editor Report · Acceptance letter]

14 Mar 2022

PONE-D-21-30492R1 

Use of a decision support system for benchmarking analysis and organizational improvement of regional mental health care: efficiency, stability and entropy assessment of the Mental Health ecosystem of Gipuzkoa (Basque Country, Spain) 

Dear Dr. Almeda:

I'm pleased to inform you that your manuscript has been deemed suitable for publication in PLOS ONE. Congratulations! Your manuscript is now with our production department. 

Kind regards, 

on behalf of

Dr. Majid Soleimani-damaneh 

Academic Editor

PLOS ONE